

# Electronic cigarettes and their association with stress, depression, and anxiety among dental students in the UAE—a pilot cross sectional study

Waseem Tawba[1], Mohamed El Qadiri[2], Mustafa Al-Adhami[2],
Nour Almehmeed[2], Natheer H. Al-Rawi[3] and Manal Awad[4]

[1] University of Sharjah, Sharjah, United Arab Emirates
[2] College of Dental Medicine, University of Sharjah, Sharjah, United Arab Emirates
[3] College of Dental Medicine, Department of Oral & Craniofacial Health Sciences, University of Sharjah, Sharjah, United Arab Emirates
[4] College of Dental Medicine, Department of Orthodontics, Pediatric and Community Dentistry, University of Sharjah, Sharjah, Sharjah, United Arab Emirates

Corresponding author
Manal Awad, awad@sharjah.ac.ae

## ABSTRACT

**Background:** During dental school, students may encounter stressful events that contribute to stress, anxiety, and depression; in response to these factors, some students use vaping or electronic cigarettes.
**Objective:** To evaluate the relationship between electronic cigarettes use and stress, anxiety, and depression among dental students.
**Methods:** A cross-sectional study included 142 dental students in their preclinical, clinical, or internship year was conducted. The 142 participants were split evenly between two groups: smokers and nonsmokers. The average age of the study's male and female participants was 21.6 years. Using the DASS 21 scale, participants were instructed to complete an electronic questionnaire assessing the association between electronic cigarettes use and stress, anxiety, and depression in smokers and nonsmokers.
**Results:** According to the findings of this study, stress, anxiety, and depression were significantly associated with electronic cigarette use. Compared to non-smokers, electronic cigarette smokers reported higher levels of *severe/extremely severe* depression (OR: 10.34, 95% CI: [4.23–24.1]), anxiety (OR: 13.8, 95% CI: [5.4–30.1]) and stress (OR: 27.6, 95% CI: [8.9–85.8]). Compared to males, females were 2.5 times (95% CI: [1.02–6.1]) more likely to report *severe/extremely severe* anxiety ($P < 0.05$).
**Conclusion:** This study demonstrates a significant correlation between the use of electronic cigarettes and elevated levels of stress, anxiety, and depression among dental students in the UAE.

## INTRODUCTION

Dental education is considered one of the most stressful experiences in undergraduate education, its intensity increases as students move on to successive years of study

(*Uraz et al., 2013*). In fact, dental education was found to be more stressful than medical and other health sciences education (*Abbasi et al., 2020*; *Murphy et al., 2009*). For example, in Denmark, *Moore, Madsen & Trans (2020)* reported that dental students scored higher than medical students on all mean test scores including the Depression Anxiety & Stress Scale (DASS-21). This high level of stress and anxiety is attributed to the demands of the dental education that requires high degree of achievement in theoretical knowledge, clinical skills and a high level of professional interaction with patients (*George et al., 2022*; *Garcia et al., 2023*). In a recent study among dental students in Malaysia, *George et al. (2022)* reported that approximately 45% of the dental students who participated in the study had moderate to extremely severe depression. Furthermore, it was recently reported that dental students who self-reported at least severe depression presented a 79% higher prevalence in history of academic failure compared to other dental students (*Garcia et al., 2023*).

It has been postulated that in general, individuals turn to smoking to relieve their symptoms of stress and anxiety (*Rajab, 2001*; *Chaiton et al., 2009*; *Perski et al., 2022*). Alternatively, continuous smoking may lead to depression or anxiety, through its effect on the pathway in the brain that regulates mood (*Chaiton et al., 2009*; *Perski et al., 2022*). Previous research established that there is a negative association between smoking and mental health (*Boden, Fergusson & Horwood, 2010*; *Fluharty et al., 2016*; *Brown et al., 2000*; *Callahan-Lyon, 2014*; *Glasheen et al., 2014*; *Lawrence, Mitrou & Zubrick, 2009*; *de Leon et al., 2002*; *Szatkowski & McNeill, 2015*). In addition, the introduction of e-cigarettes in the market in 2007, led to more studies specifically conducted to assess their health risks (*Rodakowska et al., 2020*; *Taylor et al., 2014*; *Bakhshaie, Zvolensky & Goodwin, 2015*). Furthermore, in a recent longitudinal study, *Kang & Malvaso (2024)* reported that among a UK representative sample e-cigarette use was associated with adverse general mental health, social dysfunction, & anhedonia, as well as loss of confidence. The primary active ingredient in most e-cigarette liquids is nicotine, a substance that triggers the release of dopamine a neurotransmitter linked to pleasure and reward. This can temporarily alleviate tension and anxiety symptoms, yet prolonged usage can intensify feelings of anxiety and depression over time (*Ebersole et al., 2020*).

An increasing number of young adults in the UAE are adopting this practice (*Ahmed et al., 2021*; *Abbasi et al., 2022*). In a cross-sectional study conducted by *Ahmed et al. (2021)* among students from three universities within the UAE, 15.1% acknowledged their present cigarette smoking habit, 4.1% reported using vaping or electronic-cigarettes, and 28.8% disclosed using both electronic cigarettes and traditional tobacco products.

The utilization of tobacco products, such as, conventional combustible cigarettes and novel methods of tobacco delivery, like midwakh, shisha or waterpipe, alongside various electronic nicotine delivery systems, is moving in an upward trend (*Ahmed et al., 2021*). In the UAE, the average age at which individuals initiate tobacco or nicotine usage is around 20 years old (*Abbasi et al., 2022*). Among those aged between eighteen and twenty-seven years who engage in tobacco use, 43.6% opt for electronic smoking devices.

Several studies assessed the prevalence of smoking among dental students in different countries. In which, a prevalence of smoking ranged from 17–48% (*Ahmed et al., 2021*;

*Abbasi et al., 2022*). Reasons suggested for this relatively high prevalence of smoking was related to the heavy dental course load that leads to the initiation of smoking (*Elani, Bedos & Allison, 2013*; *Smith & Leggat, 2007*; *Fujita & Maki, 2018*).

Therefore, the main objective of this study is to investigate the relationship between electronic cigarette use and its potential association with stress, anxiety, and depression among dental students in the UAE.

## METHODS

Undergraduate dental education at the University is a six-year program. The first year is a "Foundation Year" that is common to all medical and dental students. Eligibility to be admitted to the College of Dental Medicine is based on the students' performance during this preparatory year and their obtained GPA. Therefore, the level of competition is very high, and only the best students with the highest GPA are admitted to the Bachelor of Dental Surgery. Dental students enrolled in this BDS program starts clinical training in year three. The hours of clinical training increase gradually in year 4 and 5. All dental graduates must complete a 12-months internship program to qualify as dentists. A controlled cross-sectional study was conducted to assess the potential impact of electronic cigarette usage on the levels of depression and anxiety among dental students enrolled in the dental program. To enhance convenience and accessibility, the survey responses were collected digitally through an electronic format, making it simpler to reach out to dental students enrolled in the BDS program in the preclinical and clinical years. The eligibility criteria included that students must be registered full time in the undergraduate program. The survey link was widely disseminated across various social media platforms. This study was approved by the Research Ethics Committee at the University of Sharjah (REC-22-12-07-01-S).

The survey included questions regarding age, sex, and academic year. Data were collection from December 2022 to February 2023. To establish participants use of e-cigarettes, they were asked to respond "yes" or "no" to the following question "do you currently smoke e-cigarettes". The Depression, Anxiety and Stress Scale (DASS-21) measures the three dimensions of these psychological conditions in a single scale (*de Leon et al., 2002*). DASS-21 is the short-form version of the original self-reported 42-item questionnaire and has demonstrated adequate reliability and validity (*Laranjeira et al., 2023*; *Coker, Coker & Sanni, 2018*). In which the scale's was found to have high internal consistency and high discriminative, concurrent, and convergent validity (*Laranjeira et al., 2023*). DASS-21 consists of three subscales: depression subscale, anxiety subscale and stress subscale. Each subscale consists of seven items, each item scores from 0 (never), 1 (sometimes), 2 (often) to 4 (almost always), the total scores range from 0–21. The total depression subscale score that measures low self-esteem and poor outlook for the future is categorized as "normal" (0–9), "mild" (10–12), "moderate" (13–20), "severe" (21–27), and "extremely severe" (28–42). The total anxiety subscale scores that measure fear response and physiological arousal is categorized as "normal" (0–6), "mild" (7–9), "moderate" (10–14), "severe" (15–19), and "extremely severe" (20–42). The categories of the stress subscale that measures persistent arousal and tension are: "normal" (0– 10), "mild"

**Table 1 Characteristics of study participants.**

| Variable | N (%) |
|---|---|
| Gender | |
| Male | 71 (50) |
| Female | 71 (50) |
| Smoking status | |
| Smoker | 71 (50) |
| Non-smoker | 71 (50) |
| Year of study | |
| Pre-clinical years | 34 (24) |
| Clinical years | 94 (66) |
| Internship | 14 (10) |
| Age (mean ± SD) | 21.6 ± 2.1 |

**Note:**
Distribution of study participants by sex, age, smoking status and year of study.

(11–18), "moderate" (19–26), "severe" (27–34), and "extremely severe" (35–42) (*Laranjeira et al., 2023*). Noteworthy, DASS-21 is only a screening tool to reflect the severity of symptoms and should not be used as a diagnostic tool.

Data analysis was conducted utilizing the IBM SPSS software (version 27), with a focus on comparing the distribution of measured factors between the study groups. The chi-square test was employed for statistical comparisons. Additional multiple logistic regression analysis was utilized to test the association between depression, anxiety, stress and smoking status, year of study and sex. For this analysis the dependent variables were dichotomized into "severe/extremely severe" coded as "1" and "normal, mild, moderate" coded as "0". Two tailed level of significance was used (alpha level = 0.05).

## RESULTS

Table 1 presents a summary of the characteristics of the 142 students who participated in the study. The mean age of the participants was 21.6 ± 2.1 years (females: mean age 21.3 (±1.84), males: mean age 22.0 (±2.3). Among these participants, an equal number of students reported being smokers ($N = 71$) and nonsmokers ($N = 71$). In terms of the participants' academic progression, 24% ($N = 34$) were in their pre-clinical years of study, 10% ($N = 14$) were in the internship phase, while the largest group of respondents ($N = 94$ (66%)) were actively engaged in their clinical clerkship year. The mean age of the participants was calculated at 21.6 ± 2.1 years.

Significant differences were observed in the total DASS-21 subscales scores according to smoking status (smokers: stress: 24.8 (±12.8), anxiety: 21.1 (±11.6), depression: 23.5 (±13.1)) *vs.* non-smokers: stress: 10.8 (±11.3), anxiety: 8.3 (±10.4), depression: 9.5 (±11.8); $P < 0.05$) (Fig. 1).

Table 2 depicts the degrees of stress, anxiety, and depression observed in both smoking and non-smoking participants. Compared to non-smokers, a significantly ($P < 0.001$)

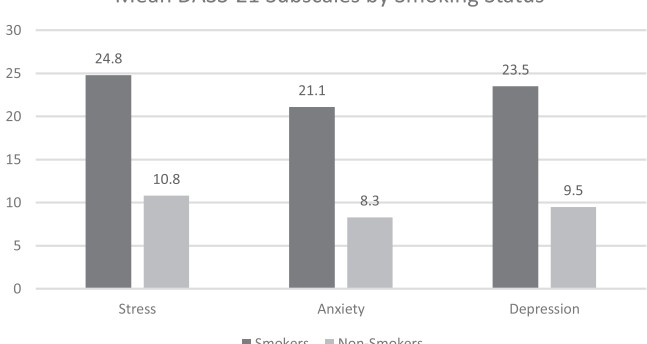

Figure 1 **Mean DASS_21 subscales by smoking status.** (A) Stress (B) Anxiety (C) Depression.

Table 2 **Levels of stress, anxiety and depression among smokers and non-smokers.**

| Stress | Smoker N (%) | Non-smoker N (%) | P value* |
|---|---|---|---|
| Normal | 16 (22.5) | 42 (59.2) | <0.001 |
| Mild | 7 (9.9) | 13 (18.3) | <0.001 |
| Moderate | 6 (8.5) | 10 (14.1) | <0.001 |
| Severe | 26 (36.6) | 2 (2.8) | <0.001 |
| Extremely severe | 16 (22.5) | 4 (5.6) | <0.001 |
| **Anxiety** | | | |
| Normal | 11 (15.5) | 45 (80.4) | <0.001 |
| Mild | 4 (5.6) | 2 (2.8) | <0.001 |
| Moderate | 7 (9.9) | 7 (9.9) | <0.001 |
| Severe | 7 (9.9) | 4 (5.6) | <0.001 |
| Extremely severe | 42 (59.1) | 13 (18.3) | <0.001 |
| **Depression** | | | |
| Normal | 15 (21.1) | 41 (57.7) | <0.001 |
| Mild | 3 (4.2) | 7 (9.9) | <0.001 |
| Moderate | 9 (12.7) | 11 (15.5) | <0.001 |
| Severe | 10 (14.1) | 5 (7.0) | <0.001 |
| Extremely severe | 34 (47.9) | 7 (9.9) | <0.001 |

Notes:
* Chi-square test.
Levels of stress, anxiety and depression among smokers and non-smokers.

higher percentage of smokers reported *severe* or *extremely severe* stress levels (8.4% *vs.* 59.1%), anxiety levels (15.9% *vs.* 69.1%) as well as depressions levels (62.0% *vs.* 16.9%).

The multivariate logistic regression analysis in Table 3 showed that smokers were significantly more likely to report *severe/extremely severe* depression (OR: 10.34 95% CI: [4.23–24.1]), anxiety (OR: 13.8, 95% CI: [5.4–35.1]) and stress (OR: 27.6 95% CI: [8.9–85.8]). Furthermore, compared to males, females were 2.5 times (95% CI: [1.02–6.1]) more likely to report *severe/extremely severe* anxiety ($P < 0.05$).

**Table 3 Association between smoking status, year of study, sex, and aspects of the DASS_21 Scale using logistic regression analysis.**

| Variable | Odds ratio | 95% CI | P value |
|---|---|---|---|
| Depression* | | | |
| Smoking | | | |
| Yes | 10.34 | [4.23–24.1] | 0.0001 |
| No* | | | |
| Preclinical years** | | | |
| Clinical years | 0.45 | [0.18–1.15] | 0.10 |
| Preclinical years** | | | |
| Interns | 0.44 | [0.10–1.75] | 0.22 |
| Sex | | | |
| Female | 1.34 | [0.59–3.1] | 0.48 |
| Male** | | | |
| Anxiety* | | | |
| Smoking | | | |
| Yes | 13.8 | [5.4–35.1] | 0.0001 |
| No** | | | |
| Preclinical years** | | | |
| Clinical years | 0.38 | [0.14–1.0] | 0.05 |
| Preclinical years** | | | |
| Interns | 0.12 | [0.02–0.55] | 0.007 |
| Sex | | | |
| Female | 2.50 | [1.02–6.1] | 0.04 |
| Male** | | | |
| Stress* | | | |
| Smoking | | | |
| Yes | 27.6 | [8.9–85.8] | 0.0001 |
| No** | | | |
| Preclinical years** | | | |
| Clinical years | 0.40 | [0.13–1.2] | 0.10 |
| Preclinical years** | | | |
| Interns | 0.23 | [0.04–1.1] | 0.07 |
| Sex | | | |
| Female | 2.34 | [0.89–6.1] | 0.08 |
| Male** | | | |

Notes:
*Dependent variable dichotomised into «severe/extremely severe compared to normal/mild/moderate.
**Reference group.

# DISCUSSION

Our findings reveal a higher prevalence of stress, anxiety, and depression among dental students, compared to studies conducted among Malaysian health science students, (*Fauzi et al., 2021*) as well as, medical students in Saudi Arabia (*Kulsoom & Afsar, 2015*). However, our findings are similar to those reported among dental students in Saudi

Arabia, Colombia and UK (*Divaris et al., 2013*; *Newbury-Birch, Lowry & Kamali, 2002*; *Basudan, Binanzan & Alhassan, 2017*). Consistent with previous research, *Magid et al. (2009)* in this study, participants who smoke e-cigarettes had significantly higher prevalence of stress, anxiety and depression compared to non-smokers. This could be explained by the self-medication theory that postulates that individuals turn to smoking to alleviate their psychological symptoms, suggesting that symptoms of depression and anxiety may lead to smoking (*Fluharty et al., 2016*; *Magid et al., 2009*; *Tien Nam et al., 2020*). The association between electronic cigarettes and heightened stress and anxiety levels can be attributed to the presence of nicotine, which has been recognized for its capacity to stimulate the hypothalamic-pituitary-adrenal (HPA) axis and the sympathetic nervous system (SNS), thereby triggering an escalation in cortisol and adrenaline release (*Tien Nam et al., 2020*). Noteworthy, the association between E-cigarette use and psychological problems may be complex, non-linear and bidirectional (*Farooqui et al., 2023*; *Lechner et al., 2017*). For example, *Lechner et al. (2017)* reported that among adolescents, depressive symptoms were associated with the initiation of e-cigarette use, and that continuous use of e-cigarettes was associated with increase in levels of depression. This association is also confounded by other external factors such as sociodemographic and economic characteristics. Therefore, a better understanding of the causal link between mental health and smoking is essential to address the determinant factors.

Our findings indicate that there is no significant difference in stress levels between students in their pre-clinical years compared to their peers who were in their clinical years or internship, after adjustment for smoking status and sex. Although these results may be regarded as counterintuitive, given that students in clinical years may experience more psychological problems due to the pressure that maybe exerted on them because they deal with patients (*Elani et al., 2014*; *Elani, Bedos & Allison, 2013*; *Alzahem et al., 2011*; *Al-Sowygh, 2013*), these findings are in agreement with previous studies (*Basudan, Binanzan & Alhassan, 2017*; *Gorter et al., 2008*; *Sugiura, Shinada & Kawaguchi, 2005*). It is possible that dental students in different years experience different stressor factors, but not necessarily higher levels. For example, students in the preclinical years could be stressed about their academic performance and their ability to continue their studies in dentistry. Once they successfully move to the clinical years, their stress and anxiety levels shift towards their clinical requirements and dealing with their patients (*Basudan, Binanzan & Alhassan, 2017*). However, to effectively address dental students' psychological issues reported in this study as well as other studies in different parts of the world, additional research is needed to further understand the causes of the psychological problems encountered.

Many studies have focused on sex difference in mental health among university students, but no consistent conclusion was drawn in this respect (*Gao, Ping & Liu, 2020*; *Adlaf et al., 2001*; *Bayram & Bilgel, 2008*; *Eisenberg et al., 2007*; *Mahmoud et al., 2012*; *Al-Qaisy, 2011*; *Wong et al., 2006*; *Gitay et al., 2019*; *Núñez-Peña, Suárez-Pellicioni & Bono, 2016*). Our findings agree with previous studies that reported higher levels of anxiety among female university students (*Bayram & Bilgel, 2008*; *Eisenberg et al., 2007*; *Mahmoud et al., 2012*) compared to their male counterparts. One possible explanation is

that emotional exhaustion prevails in females compared to males. Alternatively, women seem to be at greater risk of psychological problems, due to the combination of biological and sociocultural factors. For example, social roles may exert more pressure on females to academically succeed than males (*Gitay et al., 2019*).

Results obtained from this study highlights the importance of self-knowledge of mental health. Furthermore, based on the results of our study and findings from previous studies (*Rodakowska et al., 2020*; *Ahmed et al., 2021*; *Abbasi et al., 2022*; *Elani et al., 2014*; *Elani, Bedos & Allison, 2013*; *Smith & Leggat, 2007*; *Fujita & Maki, 2018*; *Laranjeira et al., 2023*), dental schools should consider the implementation of strategies that emphasize symptoms discovery, management of stress, anxiety and depression, as well as, reinforcement of policies against young populations' e-cigarette use. Thereby, enhancing the students' well-being, including their mental health.

### Study limitations

This study has several limitations, firstly, the cross-sectional design meant that a causal relationship between smoking and depression, anxiety and stress could not be established. Thus, longitudinal studies that take into account students' medical history, income and quality of life should be considered. In these studies the follow up of students throughout their curriculum could address the onset of smoking and the causal and temporal variation of association between smoking and stress/anxiety. Second, social desirability bias could affect the obtained results. Lastly, given the design of this study and the relatively small sample size, the findings should be cautiously interpreted because the results may not be generalizable. In addition, the study included students from one institute, which may have specific features and program setting that could affect the students' perspective. Despite these limitations, our findings underscore the association between e-cigarette use and mental health among dental students. Our results add important information to the current literature to call for dental schools to educate their students on effective strategies to manage any mental health problems they may encounter during their university years.

## CONCLUSION

This study offers initial indication of a relationship between electronic cigarette usage and heightened levels of stress, anxiety, and depression among dental students. Future research should adopt longitudinal approaches and incorporate diagnostic methods to assess both electronic cigarette use and mental health outcomes, facilitating a more comprehensive investigation into the nature of this association and the underlying mechanisms. Furthermore, interventions designed to address stress, anxiety, and depression within the dental students' population should take into account electronic cigarette usage as a plausible risk factor for mental health problems that warrants consideration.

### Funding

The authors received no funding for this work.

## Competing Interests

The authors declare that they have no competing interests.

## Author Contributions

- Waseem Tawba conceived and designed the experiments, performed the experiments, analyzed the data, authored or reviewed drafts of the article, and approved the final draft.
- Mohamed El Qadiri conceived and designed the experiments, authored or reviewed drafts of the article, and approved the final draft.
- Mustafa Al-Adhami conceived and designed the experiments, performed the experiments, analyzed the data, authored or reviewed drafts of the article, and approved the final draft.
- Nour Almehmeed conceived and designed the experiments, performed the experiments, prepared figures and/or tables, and approved the final draft.
- Natheer H. Al-Rawi performed the experiments, prepared figures and/or tables, authored or reviewed drafts of the article, and approved the final draft.
- Manal Awad performed the experiments, analyzed the data, prepared figures and/or tables, authored or reviewed drafts of the article, and approved the final draft.

## Human Ethics

The following information was supplied relating to ethical approvals (*i.e.*, approving body and any reference numbers):

The University of Sharjah Research Ethics Committee (REC-22-12-07-01-S).

## Ethics

The following information was supplied relating to ethical approvals (*i.e.*, approving body and any reference numbers):

University of Sharjah, Research Ethics Committee.

## Data Availability

The raw data is available in the Supplemental File.

## Supplemental Information

Supplemental information for this article can be found online at http://dx.doi.org/10.7717/peerj.18167#supplemental-information.

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
