# Peer review of "Electronic cigarettes and their association with stress, depression, and anxiety among dental students in the UAE—a pilot cross sectional study"

_PeerJ, doi:10.7717/peerj.18167_

## Round 0.1 · original submission · Major Revisions

Dear authors, many thanks for your submission. At this time it requires some revisions. Namely, to complete and clarify methodology. It should be the more complete and transparent possible. Please, refer to the reviewers' reports for further details (the comments from R2 are mainly in their attached PDF).

Reviewer 1 ·

Basic reporting

The manuscript is well written, language is clear and and the manuscript is well-structured
The literature references provide sufficient context, yet I would recommend adding reference to the work of Elani, H. et al. to enrich background review and to add to the discussion around the difference in stress between student in different years of the program.

See suggested referenced below:

Elani, H. W., Allison, P. J., Kumar, R. A., Mancini, L., Lambrou, A., & Bedos, C. (2014). A systematic review of stress in dental students. Journal of dental education, 78(2), 226-242.

Elani, H. W., Bedos, C., & Allison, P. J. (2013). Sources of stress in Canadian dental students: a prospective mixed methods study. Journal of dental education, 77(11), 1488-1497.

Experimental design

The methods are well described and fit the cross sectional design of this study. One thing to consider in follow up longitudinal investigations is to report the onset of smoking and the temporal variation of association between smoking and stress/anxiety

Validity of the findings

It is important to remain cautious in the interpretation of the findings of this pilot study and to indicate that these findings may not be generalizable given the limitation of this design. further investigation is needed to verify the findings.

Reviewer 2 ·

Basic reporting

Sometimes there are typing errors.
There are sufficient field provided and some suggestions were made to provide all raw data shared.

Experimental design

It's a relevant article. Methods should be described with more details but it was a suggestion.

Validity of the findings

Not all underlying data have been provided but was asked about it. Otherwise, the data are robust and statistically sound.

Annotated reviews are not available for download in order to protect the identity of reviewers who chose to remain anonymous.

Reviewer 3 ·

Basic reporting

Some important literature is ignored. There are a lot of studies about the associations between e-cigarrete use and mental health. (e.g., Anda et al., 1990 ; Bakhshaie et al., 2015 ; Bowden et al., 2011 ; Brown et al., 2000 ; Chaiton et al., 2015 ; de Leon et al., 2002 ; de Leon and Diaz, 2005 ; Flensborg-Madsen et al., 2011 ; Fluharty et al., 2016 ; Glasheen et al., 2014 ; Hagman et al., 2008 ; Lasser et al., 2000 ; Kang & Malvaso, 2024; Lawrence et al., 2009 ; Lipari and Van Horn, 2017 ; Matcham et al., 2017 ; Morris et al., 2006 ; Szatkowski and McNeill, 2015), but the authors failed to discuss them. It is important for the authors to discuss them in order to incorporate their findings to the broad literature.

Experimental design

Is it possible to introduce the validity and reliability of the measurement tool you use in the current sample?

It is largely unclear to me how you measured e-cigarette use in your research?

Validity of the findings

Without knowing how the authors assessed e-cigarette use, it is hard to judge the interpretation.

Some claims made are also not correct, there already has been longitudinal studies about e-cigarette use and mental health. The authors should correctly acknowledge it and then emphasize the novelty of their research.
https://doi.org/10.1016/j.jad.2023.11.013

---

## Round 0.2 · accepted · Accept

Dear authors,
i am happy to let you know that I am accepting your manuscript for publication in PeerJ. The office will confirm with you the necessary requirements to further publish your study. Congrats!

Reviewer 2 ·

Basic reporting

No comment

Experimental design

No comment

Validity of the findings

No comment

Reviewer 3 ·

Basic reporting

Good

Experimental design

Good

Validity of the findings

Good

Additional comments

Since the authors addressed my comments, I think it is now ready for publication